# Molecular Mechanisms of Antibiotic Resistance and Novel Treatment Strategies for *Helicobacter pylori* Infections

**DOI:** 10.3390/tropicalmed8030163

**Published:** 2023-03-11

**Authors:** Mayuna Srisuphanunt, Polrat Wilairatana, Nateelak Kooltheat, Thitinat Duangchan, Gerd Katzenmeier, Joan B. Rose

**Affiliations:** 1Department of Medical Technology, School of Allied Health Sciences, Walailak University, Nakhon Si Thammarat 80160, Thailand; 2Excellent Center for Dengue and Community Public Health, School of Public Health, Walailak University, Nakhon Si Thammarat 80160, Thailand; 3Department of Clinical Tropical Medicine, Faculty of Tropical Medicine, Mahidol University, Bangkok 10400, Thailand; 4Hematology and Transfusion Science Research Center, School of Allied Health Sciences, Walailak University, Nakhon Si Thammarat 80160, Thailand; 5Akkhraratchakumari Veterinary College, Walailak University, Nakhon Si Thammarat 80160, Thailand; 6Department of Fisheries and Wildlife, Michigan State University, East Lansing, MI 48823, USA

**Keywords:** *Helicobacter pylori*, antibiotic resistance, salvage therapy, drug development

## Abstract

*Helicobacter pylori* infects approximately 50% of the world’s population and is considered the major etiological agent of severe gastric diseases, such as peptic ulcers and gastric carcinoma. Increasing resistance to standard antibiotics has now led to an ever-decreasing efficacy of eradication therapies and the development of novel and improved regimens for treatment is urgently required. Substantial progress has been made over the past few years in the identification of molecular mechanisms which are conducive to resistant phenotypes as well as for efficient strategies to counteract strain resistance and to avoid the use of ineffective antibiotics. These involve molecular testing methods, improved salvage therapies, and the discovery of novel and potent antimicrobial compounds. High rates of prevalence and gastric cancer are currently observed in Asian countries, including Japan, China, Korea, and Taiwan, where concomitantly intensive research efforts were initiated to explore advanced eradication regimens aimed at reducing the risk of gastric cancer. In this review, we present an overview of the known molecular mechanisms of antibiotic resistance and discuss recent intervention strategies for *H. pylori* diseases, with a view of the research progress in Asian countries.

## 1. Introduction

Since its initial discovery, *H*. *pylori* became the most intensively characterized microbial pathogen, mainly because of the unique infection site, the link to severe gastric diseases, and the extraordinary overall prevalence rate, as it is estimated to affect 50% of the world’s population [1]. *H*. *pylori*, a gram-negative, microaerophilic, and extremophile microorganism, causes serious gastric disorders, such as gastric atrophy, peptic ulcer disease (PUD), gastric adenocarcinoma, and gastric cell lymphoma (mucosa-associated lymphoid tissue, MALT), which renders *H*. *pylori* the only bacterial pathogen which is classified as a class I carcinogen [2,3]. *H*. *pylori* possesses a wide array of virulence factors that assist in establishing a continuous infection and ascertain bacterial survival under acidic conditions in the stomach [4]. As infected individuals can remain symptomless over long periods, prevalence rates are most likely underdiagnosed—in particular, in developing countries with relatively poor public health infrastructure [5]. It is noteworthy that, albeit infection rates are seemingly extremely high, only a small fraction (less than 1%) of infected individuals develop severe gastric complications. Nevertheless, the high incidence of *H*. *pylori* infections suggests an increasing burden on public health care, especially in developing countries where infections during early childhood appear to occur more frequently than in developed countries [6]. In South-East Asian countries, gastric cancer is often diagnosed in the advanced stages of the disease with a poor prognosis for 5-year survival [7]. Considerable differences in H. pylori prevalence are observed for different regions of the world, and the large variations in prevalence have been attributed to dietary behavior, water quality, and socioeconomic conditions [8]. 

Given the fact that *H*. *pylori* represents the primary risk factor for gastric cancer (GC) and the cause of cancer-related death, immense resources are currently dedicated to the treatment and eradication of *H*. *pylori* infections. For decades, a triple therapy—consisting of a proton-pump inhibitor (PPI: omeprazole, pantoprazole) and 2 antibiotics (usually clarithromycin combined with either metronidazole, a synthetic nitroimidazole, or ampicillin)—served as first-line therapy [9]. As resistance rates to clarithromycin and metronidazole increased to unacceptable levels, clarithromycin-resistant *H*. *pylori* was included by the WHO, in 2017, in the list of high-priority, antibiotic-resistant bacteria (Figure 1) [10]. The triple therapy is now progressively being replaced in areas of high clarithromycin resistance rates by a bismuth-containing, “three-in-one”, quadruple therapy (BQT) [11,12].

The excessive use of antibiotics for the chemotherapy of *H*. *pylori* infections has inevitably created a situation in which resistance development exceeded alarming levels and extensively compromised the usefulness of these antibiotics for the eradication of H. pylori. The declined efficacy has now incited intensive efforts for the discovery not only of novel antimicrobial compounds but also for the development of innovative regimens for treatment [14]. Despite rapid economic growth over the past decades, countries in Asia still face a deteriorating situation of *H*. *pylori* infections [15]. On the other hand, there has been tremendous progress made in this region with research efforts aimed at better control of H. pylori-related diseases that have resulted in an impressive and ever-increasing number of publications.

This review aims to present a condensed view of current progress in the field from an Asian perspective with a view of resistance mechanisms and strategies to surmount them. For a comprehensive overview, the reader is directed to several excellent reviews summarizing key aspects and current challenges of *H*. *pylori* antimicrobial therapy [16]. We apologize to authors whose work we have failed to cite owing to space constraints.

## 2. Epidemiology

Eradication strategies for *H*. *pylori* infections are currently being massively challenged by increasing antibiotic resistance, which can ultimately result in treatment failure and unclear therapeutic outcomes [16]. Strains of *H*. *pylori* can exhibit considerable resistance to the most widely used antimicrobial drugs: clarithromycin, levofloxacin, metronidazole, amoxicillin, and tetracycline (Figure 2). The antibiotic resistance of *H*. *pylori* is commonly assessed using the epsilometer test (E-test), a commercially available, culture-based method using a predefined gradient of antibiotic concentrations on a plastic strip to determine the minimum inhibitory concentration (MIC) of antibiotics [17]. The European Committee on Antimicrobial Susceptibility Testing (EUCAST) has issued a recommendation to consider MIC values > 0.5 mg/L for clarithromycin and >8 mg/L for metronidazole as indicators of H. pylori strain resistance [18]. However, it should be noted that MICs for resistant strains can display considerable variations between 0.5 to 256 mg/L for clarithromycin and 0.8 to 8 mg/L for metronidazole [19]. Marked geographic differences are also observed for resistance rates to specific antibiotics. While in Brazil and Germany, resistance rates to clarithromycin were 28.7% and 23.2%, respectively, they were only 7.3% in Iran and 0% in Malaysia [18]. Earlier data for H. pylori antibiotic resistance and resistance trends over a 6-year period of surveillance can be found in Ghotaslou et al. [20].

A survey among 9 ASEAN countries (Thailand, Cambodia, Indonesia, Laos, Malaysia, Myanmar, Philippines, Singapore, and Vietnam) was conducted to evaluate the prevalence of infection and resistance rates against metronidazole, amoxicillin, and clarithromycin [21]. *H*. *pylori* prevalence can vary considerably, ranging from 20% in Malaysia, 21–54% in Thailand, and 69% in Myanmar, while resistance to clarithromycin was highest in Cambodia (43%) and lowest in the Philippines (2%) and Myanmar (0%); however, data for the latter may reflect inadequate surveillance of resistance due to poor public health infrastructures. It is further mentioned that, at the time of the survey in Laos and Cambodia, culture-based methods for susceptibility testing were not available. Resistance to metronidazole is relatively common, whereas resistance to amoxicillin was found to be rare. The wide range of resistance rates between the different countries has urged the need for susceptibility testing prior to treatment. Current prevalence and antibiotic resistance patterns were analyzed in a 4-year retrospective study conducted from 2013 to 2017 at King Chulalongkorn Memorial Hospital in Thailand [22]. Overall, 92.61% of 1258 patients responded to initial treatment, while the remaining 95 patients responded to second-line treatment with higher doses or different antibiotics. H. pylori strains with resistance to ciprofloxacin, metronidazole, and clarithromycin were observed at rates of 21.43, 14.29, and 10.71%, respectively, whereas no strains resistant to amoxicillin, tetracycline, or levofloxacin were found. An interesting observation by the authors even suggested declining resistance rates for amoxicillin, metronidazole, levofloxacin, and tetracycline.

## 3. Molecular Mechanisms of Antibiotic Resistance 

Mutational changes leading to genetically modified drug targets appear to represent the most common type of resistance mechanism. Nevertheless, it appears that mutations affecting membrane permeability, biofilm development, and efflux pump systems are becoming more and more significant for the manifestation of drug-resistant genotypes [23].

Amoxicillin has been used as a standard drug for the treatment of *H*. *pylori* infections for decades and was initially considered “resistance-resistant” since very few examples of strain resistance had been discovered by then [24]. Later, it turned out that high rates of resistance (49.5 to 72.7%) to amoxicillin can frequently arise, in particular, as a result of unsuccessful eradication attempts [25]. Resistance to amoxicillin occurs predominantly through mutations in the PBP gene and changes in membrane permeability. Primary resistance rates for amoxicillin (3 to 5%) are relatively small, and they are considerably lower than those for metronidazole and clarithromycin [26]. Nevertheless, resistance to amoxicillin was identified as an independent risk factor for treatment failure of the standard triple therapy at different breakpoints. Resistance rates were estimated to be approximately 11% at a breakpoint MIC > 0.125 mg /L [26]. It was speculated that increased rates of amoxicillin resistance correlate with the unregulated use of the antibiotic in Asia and Southern America, where amoxicillin is available without prescription [20].

*H*. *pylori* strains resistant to clarithromycin, an inhibitor of bacterial protein biosynthesis, usually carry point mutations in the 23S rRNA of the 50S ribosomal subunit. A recent study conducted at Peking University has identified genetic determinants of antibiotic resistance against clarithromycin and levofloxacin through high-throughput nucleotide sequencing [27]. Mutation sites related to clarithromycin were identified as peptidyl transferase in the V domain of 23S rRNA, while *gyrA*, a member of the DNA topoisomerase type II family, was related to levofloxacin resistance. The most common mutant sites in clarithromycin-resistant gene sequences were A2143G and A2142G. The most frequent levofloxacin resistance mutations were N87K, D91N, and D91G. A fraction (13.5%) of the investigated isolates had double resistance mutations for both clarithromycin and levofloxacin.

Aside from the A2142G and A2143G mutations, a study recently performed in Sudan revealed the existence of a T2182C mutation within the V domain of the 23S rRNA gene [28]. The authors did not observe any associations of 23S rRNA point mutations with gender, age group, or patients’ geographical location. However, the number of sequenced samples (25) was relatively small. The screening of 169 patients positive for chronic gastritis in Vietnam, using PCR-RFLP, found the A2143G mutation in 36.1% of samples and the A2142G mutation in 3.6%; however, it failed to detect the A2142C mutation [29].

A novel levofloxacin-resistance mutation in *gyrA*, consisting of the insertion of five amino acid residues (QDNSV) immediately after the start codon, and a substitution mutation at R295H were identified through the whole-genome level sequencing of 38 clinical isolates from patients in Karnataka, India who had been diagnosed with gastritis, peptic ulcer disease, or intestinal metaplasia [30]. The authors concluded that, in light of these results, even moderate resistance to metronidazole and levofloxacin could lead to treatment failure; therefore, they have proposed a triple therapy utilizing amoxicillin, tetracycline, and PPI as an alternative first-line treatment regime. Double mutations (N87T, D91N) and single mutations (N87I and N87T) were identified in the *gyrA* gene of isolates obtained from the gastroesophageal mucosa, whereby the N87I mutation produced high resistance to levofloxacin at a MIC ≥ 32 μg/mL [31]. Mutations in the *gyrB* gene seem to occur rarely, and the mutation S479G and at position 463 were recently identified [31,32].

An important mechanism of *H*. *pylori* drug resistance is the reduction of drug influx by structural modifications of the lipopolysaccharide (LPS) membrane component. The recently described *rfaF* gene (previously *waaF*) functions as heptose transferase in the LPS core biosynthesis pathway, whereby mutations are conducive to the “deep coarse lipopolysaccharide” phenotype, which decreases the drug-permeability of the cell membrane [33]. This phenotype produces strains that are cross-resistant to amoxicillin, tetracycline, and clarithromycin, whereas only marginal resistance to chloramphenicol was associated with this phenotype [34]. It was observed that the *rfaF* gene of drug-resistant clinical strains displayed high mutation rates, with the K331R mutation being the most frequent (44.44%), a finding which makes the *rfaA-*encoded enzyme a promising candidate for evaluation as a potential drug target.

Upregulated efflux pump activity, owing to an increased expression of tolC homologous genes (*hefA*), was characterized using real-time PCR with clarithromycin- and metronidazole-resistant strains isolated from patients with gastroduodenal disorders in Iran [35]. A small portion (9.5%) of the investigated strains were multidrug-resistant *H*. *pylori* (MDR) strains with resistance against metronidazole and clarithromycin. A sequence analysis of the *rdxA* and *frxA* genes of the metronidazole-resistant strains and the 23S rRNA for the clarithromycin-resistant strains was performed to determine the genetic modifications leading to drug resistance. The most frequently occurring mutations were within the *rdxA* gene (85.5%) and the A2143G point mutation in the 23S rRNA (63.1%). Both mutations were also present in the MDR strains. The *rdxA* gene encodes an oxygen-insensitive, NADPH-dependent nitroreductase, which was associated with metronidazole resistance in earlier reports [36]. It was proposed that mutational inactivation of *rdxA* and other reductase-encoding genes, such as *frxA* (encoding a NAD(P)H-flavin oxidoreductase) and *fdxB* (ferredoxin-like protein), would be favorable to the formation of the resistant phenotype [37]. *H*. *pylori* strains resistant to metronidazole and levofloxacin, analyzed using whole-genome sequencing, were shown to express non-functional or altered *RdxA* and/or *FrxA* proteins resulting from nonsense or frameshift mutations in the coding sequences and containing partial gene deletions [30]. However, the function of the *rdxA* gene product, as well as the related *frxA* gene, is still subject to debate. Allelic replacement of wild-type *rdxA* with truncated *rdxA* resulted in metronidazole resistance, whereas replacement with missense-mutated *rdxA* did not result in detectable resistance [38]. It was suggested that resistance to metronidazole can arise in *H*. *pylori* without mutations in *rdxA* or *frxA*, thus supporting the notion that other genetic elements are likely involved in metronidazole resistance, a finding corroborated by a recent report demonstrating that metronidazole-resistant strains can carry intact genes for both oxidoreductases [30]. It is noteworthy that, mainly because of the accelerating resistance development attributed to high prescription rates, fluoroquinolones are now infrequently used as first-line treatment drugs for *H*. *pylori* infections; however, they still may have some effectiveness as second-line drugs [39].

Several observations in the literature suggest a relationship between specific genotypes for virulence factors and the degree of gastroduodenal diseases [40]. The absence of the *cagA* genotype was proposed to be linked to the development of metronidazole resistance [41]. *VacA* is an important virulence factor with pleiotropic effects involved in gastric mucosa colonization and disease progression [42]. The *vacA* gene encodes several polymorphic regions (s, m, i), and clarithromycin resistance was significantly linked to the *vacA* i-allele, whereby the clarithromycin-resistance mutation, A2142G, was 3-fold more frequent in *vacA* i1 strains than *vacA* i2 strains [43]. Resistance to clarithromycin, metronidazole, and amoxicillin was found to be increased in strains harboring *cagE* and *vacA* s1a/m2 genotypes [44]. Wang et al. have attempted to correlate resistance to the five most commonly used antibiotics (metronidazole, levofloxacin, clarithromycin, amoxicillin, and tetracycline) to the presence of genes encoding virulence factors. Clarithromycin resistance was associated with *iceA*, while resistance to metronidazole was related to *vacA*; levofloxacin resistance was concerned with *cagA* and *slyD,* and amoxicillin resistance was associated with *iceA* [45].

## 4. Rescue Therapy

Eradication confirmation 4–6 weeks post-treatment is commonly used to assess the success of the therapeutic efforts. Failure of initial treatment can occur in up to 20% of patients, and while patient-specific factors may play a role to some extent, major complications and, in particular, re-infection can arise due to poor medication compliance [46]. For this situation, salvage therapies have been developed that renounce the use of first-line antibiotics, especially clarithromycin and metronidazole. Instead, bismuth quadruple therapy (BQT) using a PPI, bismuth subcitrate or subsalicylate, amoxicillin, and tetracycline or levofloxacin triple therapy (PPI, levofloxacin, and amoxicillin) are frequently considered as alternative treatment options. Regional antibiotic resistance profiles can be analyzed using molecular susceptibility testing, which improves treatment success rates.

Rifabutin-based triple therapy has recently emerged as a promising alternative to conventional treatment with clarithromycin and metronidazole [47]. *H*. *pylori* is highly susceptible to rifabutin, a derivative of rifamycin and an inhibitor of the prokaryotic RNA polymerase. Marketed under the brand name ‘Talicia’, the therapy consists of two antibiotics, amoxicillin and rifabutin, and the PPI omeprazole. In patients with confirmed adherence to treatment, eradication rates can be as high as 90%, and reported resistance development has been minimal when compared to standard drugs [48,49].

A therapeutic regimen designated as LOAD (levofloxacin, omeprazole, Alinia, and doxycycline) has gained some attention as a second-line therapy, especially in cases of BQT failure [50]. Alinia is the brand name for nitazoxanide, which was originally discovered as a broad-spectrum antiparasitic drug. However, the antimicrobial activity spectrum resembles that of metronidazole, making this drug a potential alternative to metronidazole in *H*. *pylori* eradication regimens [51]. Three enzymes, including a pyruvate oxidoreductase, were identified as mediators of susceptibility to nitazoxanide in *H*. *pylori* strains [52]. Notably, no clinically significant levels of resistance were observed during clinical studies or during long-term in vitro exposure of *H*. *pylori* strains to the drug [53].

A modification of the quadruple therapy, comprised of furazolidone, amoxicillin, bismuth, and a PPI, was recently explored in a clinical setting [54]. Furazolidone is a monoamine oxidase inhibitor and nitrofurantoin-type antibiotic commonly used in Asia. The furazolidone-based therapy demonstrated high eradication rates, exceeding 90%, and was found to be suitable as a first-line treatment for *H*. *pylori* infection in areas with a high prevalence of clarithromycin resistance. The clinical usefulness of this therapy was further supported by the fact that adverse effects were mild and occurred at a low incidence.

While not an antibiotic by itself, the antisecretory drug vonoprazan (Takecab) a potassium-competitive acid blocker (P-CAP), has emerged as an acid-suppressing agent with a much higher (~350-fold) potency than standard PPIs [55]. Unlike conventional PPIs, vonoprazan is a reversible H+-K+ ATPase inhibitor. Eradication rates >90% were observed in a study of patients receiving triple therapy with vonoprazan, amoxicillin, and clarithromycin or metronidazole [56]. It is, therefore, conceivable that vonoprazan could contribute to conquering the increasing prevalence of antibiotic resistance associated with treatment regimens including conventional PPIs for the eradication of *H*. *pylori* infections.

## 5. Antimicrobial Susceptibility Testing

The identification of mutations conducive to resistance genotypes using clinical biopsy specimens would ideally represent a strategy for clinicians to find and adjust efficient treatment options. Antimicrobial susceptibility testing allows for the identification of resistant strains of *H*. *pylori* and excludes the use of ineffective antibiotics, especially in regions with high antibiotic resistance [57]. In addition to standard bacterial culture methods, rapid, culture-independent molecular assay formats exist that allow for the identification of resistance mutations and the assessment of the efficacy of the intended therapy. These include the PCR-RFLP detection of point mutations [58], droplet digital PCR [59], the amplification refractory mutation system (ARMS-PCR) [60], and test kits, such as GenoType® HelicoDR (Hain Lifescience GmbH, Nehren, Germany) [61], which allow for the simultaneous detection of resistance mutations to clarithromycin and fluoroquinolones [62]. Dual-priming, oligonucleotide-based multiplex polymerase chain reaction (DPO-PCR) has been used in tailored first-line therapies in South Korea with satisfactory results [63].

The disadvantage of these procedures is the requirement of primers, fluorescent probes, and specialized equipment that is not available in most clinical settings. It is also noteworthy that, whereas two prominent sites of mutation in the 23S rRNA are recognized for clarithromycin resistance, metronidazole resistance involves several mutations in *rdxA* and other genes that are more difficult to detect.

The tailored therapy has demonstrated its practical usefulness, particularly in cases of second-line or third-line refractory infections in combination with PCR-based molecular tests [64,65]. The surveillance of strain resistance using susceptibility testing can now be carried out with stool samples, thus obviating the need for invasive methods, such as endoscopy (stool polymerase chain reaction) [66]. 

Of particular interest is the question of whether the detection of a resistant genotype is concurrently correlated with the existence of a drug-resistant phenotype. Zhang et al. have compared the results of ARMS-PCR to the E-test MIC drug sensitivity assay for clarithromycin and reported a statistically significant correlation (*p* > 0.05, area under the receiver operating curve = 0.969) between the two methods [60]. However, some strains, tested as wild-type using ARMS-PCR, appeared to be drug-resistant, thus suggesting that the clarithromycin-resistant phenotypes may not be limited to sites 2142 and 2143, but may carry mutations at different positions.

## 6. Novel Treatment Options

Increasing resistance rates observed for the most commonly used antibiotics against *H*. *pylori* infections have now stimulated intensive efforts to discover novel antimicrobial compounds and leads that are useful for structure-based drug design [67]. The conventional, alternative, and novel treatment options for *H*. *pylori* are summarized in Table 1.

A recent study from China has described the potent antimicrobial activities of armeniaspirol A (ARM1) against MDR strains of *H*. *pylori* [68]. Armeniaspirols are antibiotics containing an unusual spiro(4.4)non-8-ene moiety bio-synthesized by Streptomyces armeniacus [69]. The compound has membrane-disrupting properties, which ultimately lead to an inhibition of biofilm formation. Remarkably, dual therapy with ARM1 and omeprazole demonstrated bactericidal effects comparable to the standard triple therapy in a mouse model of MDR H. pylori strains, while concurrent toxicity against normal tissues was found to be negligible [68]. The bactericidal activity of ARM1 appeared to be significantly higher than that of metronidazole. Similar to armeniaspirol, the natural herb compound dihydrotanshinone I (DHT) also exhibited antibacterial activity through the elimination of preformed biofilms and biofilm-encased *H*. *pylori* cells [70]. DHT showed strong time-dependent bactericidal activity, with MIC50/90 values of 0.25/0.5 µg/mL.

A promising opportunity for pharmacological intervention is the inhibition of the response stimulator HsrA, an electron-transfer flavodoxin protein that is essential for numerous metabolic functions [71,72]. A screening of 1120 compounds contained in an FDA drug library resulted in the identification of seven natural flavonoids that inhibited the DNA-binding activity of HsrA. Phytochemicals, chrysin, and galangin exhibited marked synergistic bactericidal activity in combination with clarithromycin or metronidazole [73,74,75]. Investigations of flavodoxin inhibitors have also been extended to the evaluation of novel synthetic nitroethylene- and 7-nitrobenzoxadiazole-based inhibitors of *H*. *pylori* flavodoxin, showing promising therapeutic indexes [76]. Oral administration to a mice model of *H*. *pylori* infection revealed low toxicity and decreased rates of gastric colonization. However, at present, detrimental effects on the gut microbiota remain to be investigated in greater detail.

**Table 1 tropicalmed-08-00163-t001:** Conventional, alternative, and novel treatment options for *Helicobacter pylori*.

Substance	Type	Mechanism of Action	Usage in Treatment	References
Amoxicillin	Antibiotic	Inhibition of cell wall biosynthesis	Conventional	[26,54,77]
Clarithromycin	Antibiotic	Inhibition of bacterial protein synthesis	Conventional	[28,35,60]
Levofloxacin	Antibiotic	Inhibition of bacterial DNA synthesis	Conventional	[39,78,79]
Metronidazole	Antibiotic	Inhibition of protein synthesis via DNA structure and strand breakage	Conventional	[35,52,72]
Tetracycline	Antibiotic	Inhibition of protein synthesis via the inhibition of mRNA-ribosome complex formation	Conventional	[75,80,81]
Rifabutin	Antibiotic	Inhibition of bacterial RNA polymerization	Alternative, Combination	[47,48,49]
Doxycycline	Antibiotic	Inhibition of protein synthesis via the inhibition of mRNA-ribosome complex formation	Alternative, Combination	[50,77,78]
Nitazoxanide	Antibiotic	Interfering with anaerobic energy metabolism	Alternative	[51,52,53]
Furazolidone	Antibiotic	Inhibition of protein synthesis via DNA cross-linkage	Clarithromycin-, Metronidazole-resistant	[54,73,74]
Armeniaspirol A	Antibiotic	Disruption of the bacterial cell membrane, inhibition of biofilm formation	Novel Alternative	[68,69,82]
Dihydrotanshinone I	Phytochemical	Elimination of preformed biofilm	Novel Alternative	[70,83]
Chrysin	Phytochemical	Interfering with cell wall formation, vesicle formation, and cell lysis	Novel Alternative	[71,84,85]
Galangin	Phytochemical	Interfering with cell wall formation, vesicle formation, and cell lysis	Novel Alternative	[71,84,85]
Curcumin	Phytochemical	Inhibition of vacuolation via binding to the virulence factor	Novel Alternative	[86,87,88]
Pexiganan	Peptide	Binding to the bacterial membrane, forming a toroidal pore	Novel Alternative	[89,90,91]
Tilapia Piscidins	Peptide	Induction of membrane micelle formation	Novel Alternative	[90,92]
Epinecidin-1	Peptide	Generation of membrane curvature, vascularization, and pore formation	Novel Alternative	[90,92,93]
Cathelicidins	Peptide	Shrinking of the flagella and pore formation on bacterial membrane	Novel Alternative	[90,94]
Defensins	Peptide	Permeabilization of bacterial cell membrane	Novel Alternative	[90,95,96]
Bicarinalin	Peptide	Permeabilization of bacterial cell membrane	Novel Alternative	[90,97]
Odorranain-HP	Peptide	Unclear	Novel Alternative	[90,98]
PGLa-AM1	Peptide	Binding to the bacterial cell membrane	Novel Alternative	[90,99]
Bacteriocins	Peptide	Binding to the bacterial cell membrane, pore formation	Novel Alternative	[90,100,101]

Curcumin (diferuloylmethane) and polyphenolic plant metabolites display a multitude of anti-microbial, anti-inflammatory, and anti-proliferative properties and are currently being explored as potential therapeutic candidates against *H. pylori* [88,102]. Curcumin has not only demonstrated pronounced antimicrobial effects in *H. pylori* infected C57BL/6 mice but also seems to contribute to the reduction or repair of gastric epithelium damage, as revealed by histological analysis [103]. The cytotoxin-associated gene A protein (*CagA*) is involved in the establishment of a persistent *H. pylori* infection, the malignant transformation of gastric cells, and consequently, is a major factor in the development of gastric cancer [104]. The interaction of *CagA* with curcumin and its metabolites has been proposed to contribute to the suppression of *CagA* oncogenic activity [105]. It is important to note that curcumin failed, in a large number of studies, to demonstrate clinically useful effects; nevertheless, the possibility exists that despite their low solubility and poor adsorption and bioavailability, optimized derivatives of curcumin, such as EF24, may prove useful as supplementary agents for the treatment of *H. pylori* infections [106,107].

Antimicrobial peptides (AMPs) with activity against drug-resistant *H. pylori* were recently identified in an Iranian study through a screening of PubMed, Scopus, and Web of Science databases [90]. Nine groups containing 22 antimicrobial peptides were comprised of pexiganan, tilapia piscidin, epinecidin-1, cathelicidins, defensins, bicarinalin, odorranain-HP, PGLa-AM1, and bacteriocins—whereby the highest anti-H. pylori activities were observed for pexiganan, tilapia piscidin, and PGLa-AM1. The latter represents five kDa peptides with a predominant α-helical structure, cationic charge, and high isoelectric points. The 22-amino-acid residue peptide, pexiganan, which is a magainin AMP analog, was incorporated into chitosan-alginate polyelectrolyte complex pexiganan nanoparticles (PNPs) with a particle size of 415 ± 26 nm for enhanced delivery of the AMP to the gastric mucosa [108]. The results confirmed that PNPs efficiently eradicated H. pylori in the mouse stomach and showed improved peptide stability and prolonged retention times. Moreover, the strategy of mucoadhesive delivery of nano- or microparticles offers the benefit of effective drug penetration across the mucus layer, increased resistance against proteolytic degradation, and would also protect acid-sensitive drugs from fast degradation.

While the use of genetically engineered symbiotic lactic acid bacteria as carriers for AMP delivery may constitute a future treatment option for *H. pylori*, probiotics were intensively studied as part of *H. pylori* treatment regimens [109]. Probiotics display a wide range of antibacterial effects, including the secretion of organic acids and bacteriocins, the formation of H2O2, the inhibition of adhesion processes, and the downregulation of proinflammatory factor expression [110].

The results of earlier studies suggest that specific probiotics, such as *Saccharomyces boulardii* and *Lactobacillus johnsonni* La1, probably diminish the bacterial load, but do not completely eradicate the *H. pylori* bacteria when administered as monotherapy [109]. Adjuvant therapy, employing combinations of probiotics and antibiotics, appears to be more effective, as the probiotics contribute significantly to diminishing the iatrogenic side effects of the drug treatment. Inhibitory effects on *H. pylori* growth and a significant reduction in antibiotic-associated side effects were observed with the novel preparation of *Lactobacillus reuteri* [111]. *Lactobacillus reuteri* synthesizes a non-peptidic antipathogen compound, reuterin, reported to downregulate the genes for virulence factors *vacA* and *flaA* [112]. These findings have confirmed various beneficial effects of probiotics not only for *H. pylori* infections but also for several other gastrointestinal diseases.

Herbal therapy for gastric diseases has a long history of demonstrated gastroprotective and antimicrobial effects in *H. pylori* infections [113]. Therapeutic strategies for the effective treatment of *H. pylori* infections are now being broadly augmented through increasing research efforts aimed at the clinical evaluation of herbal and traditional Chinese medicine. An overview of the rapidly expanding field is presented by Li Y et al. [114]. The current knowledge about phytomedicines used for *H. pylori*-associated gastric diseases is summarized by Salehi et al. [115]. Innovative solutions to overcome current treatment deficits, such as resistance, could conceivably arise from integrated Chinese and Western medicine [116,117]. Aside from their use as an herbal cure, per se, phytochemicals that have been identified in numerous studies may offer prospects for the discovery of lead compounds for the design of novel and synthetic therapeutic agents [118].

The emergence of the antimicrobial resistance of *H. pylori* to important antibiotics keeps increasing, and antimicrobial resistance (AMR) is risky if it leads to treatment failure clinically and (gastroduodenal) histopathologically. However, there have been no previous reports specifying clinical status (e.g., higher prevalence rates of peptic ulcer, carcinoma, or lymphoma) if there is treatment failure due to antimicrobial resistance. The previous reports have only mentioned that antimicrobial resistance was associated with treatment failure [119,120,121]. In individual patients, mechanisms of resistance deployed by *H*. *pylori* cause treatment failures, diagnostic difficulties, and ambiguity in the clinical interpretation of therapeutic outcomes. At the population scale, globally increasing antibiotic resistance has led to a substantial decrease in efficacy of *H. pylori* treatment and probably an increased risk of complications, such as peptic ulcers and gastric cancer [16,122]. This resistance becomes more important in areas or countries where *H. pylori* has high resistance or multi-resistance to the antibiotics used in treatment regimens. Since the regimens may have a high risk of clinical or histopathological treatment failure, the drug sensitivity of *H. pylori* should be regularly reported in highly antibiotic-resistant countries [119,120]. As a result, WHO has declared that AMR is a global health and development threat. It requires urgent multisectoral action in order to achieve the Sustainable Development Goals (SDGs) [118,123].

Taken together, intensive research efforts and the development of novel and enhanced therapeutic options in the ‘post-antibiotic’ era have generated promising alternatives and treatment strategies to combat infections with drug-resistant *H*. *pylori*.

## 7. Conclusions

The deteriorating problem of antibiotic-resistant *H*. *pylori* is now being targeted by a multitude of research efforts aimed at understanding the molecular mechanisms of resistance and the development of novel and alternative options for treatment. Research in Asian countries has greatly contributed to the wealth of data accumulated over the previous years. Much progress has been made in the molecular analysis of genetic alterations leading to drug resistance, in particular, for the frequently applied antibiotics clarithromycin and metronidazole. Culture-independent molecular methods now allow for the implementation of susceptibility-guided therapies adapted to local resistance profiles, with a largely reduced risk of treatment failure. Efficient rescue therapies using less resistance-prone, second-line drugs, such as rifabutin, amoxicillin, and furazolidone, can be used to achieve high eradication rates. Screening for novel antimicrobial compounds has identified promising candidates for drug development, such as flavodoxin inhibitors, antimicrobial peptides, antibiotics, and phytochemicals traditionally used in Chinese medicine, which warrant further investigation. It can be concluded that, despite the clinical challenges, gastrointestinal diseases caused by *H*. *pylori* can, at large, be efficiently managed using appropriate diagnostic tools, improved regimens for treatment, and post-treatment confirmation of eradication.

## Figures and Tables

**Figure 1 tropicalmed-08-00163-f001:**
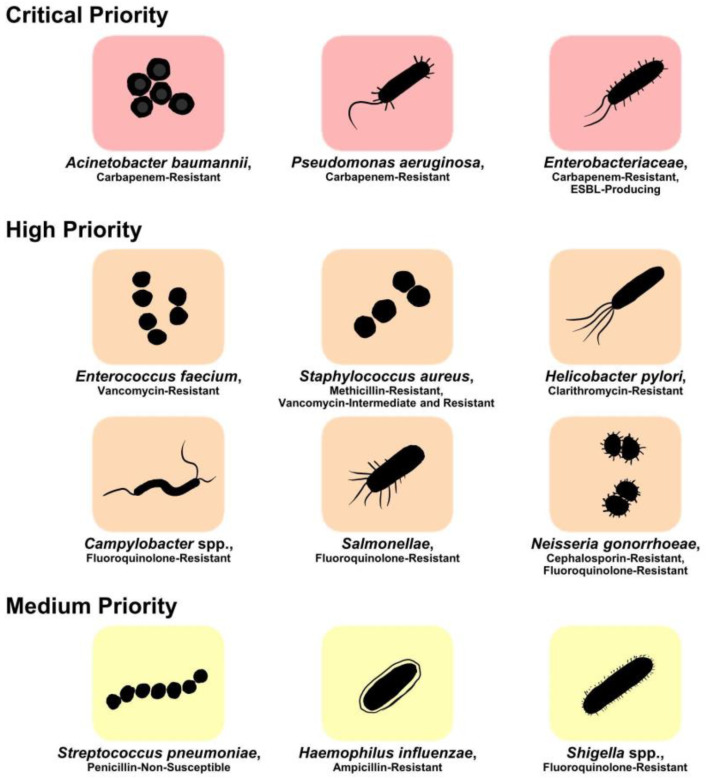
WHO global priority list of antibiotic-resistant bacteria to guide research, discovery, and development of new antibiotics, adapted with permission from WHO [13]. *Helicobacter pylori* is listed as a high-priority organism.

**Figure 2 tropicalmed-08-00163-f002:**
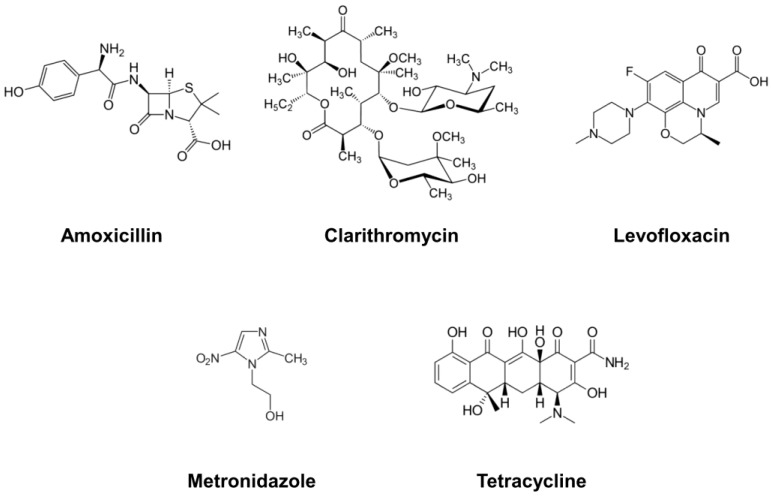
Structural formulas of the antibiotics used most frequently for the treatment of *H*. *pylori* infections. Amoxicillin, clarithromycin, levofloxacin, metronidazole, and tetracycline are shown. Their use has resulted in the rise of resistant strains, rendering therapeutic approaches increasingly problematic.

## Data Availability

Not applicable.

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
