# Peer review of "Molecular Mechanisms of Antibiotic Resistance and Novel Treatment Strategies for Helicobacter pylori Infections"

_tropicalmed, 2023, doi:10.3390/tropicalmed8030163_

Round 1
Reviewer 1 Report
The authors made a interesting review based on evidence, they refer properly the author for material they cited, Besides they point some important issues out of resistant. I would like to read regarding the clinical outcome more than countries isolation! I t would be better if they point out this issue, justo to know if the antimicrobial resistant is higher associated to specific clinical status. Besides, the action over virulence factors. As well as other important variables the authors can selected to give more specific view where the resistant become more important.
Author Response
Response to Reviewer 1 Comments : 2th Revision
Question 1. If antimicrobial resistant is higher associated to specific clinical status?
Response 1: We have now replied Reviewer 1 comments and added the messages at line 389-393, followed the addition reply as below:
CONSEQUENCES OF HELICOBACTER PYLORI RESISTANCE TO ANTIBIOTICSAntibiotic resistance is important if it leads to treatment failure. However there is no report to clearly specify clinical status if there is antimicrobial resistant.
Antimicrobial resistance is risky to H. pylori treatment failure clinically and (gastroduodenal) histopathologically. However there has been no previous report to specify clinical status ( eg. higher prevalence of peptic ulcer, carcinoma or lymphoma) if there is treatment failure due to antimicrobial resistance. The previous reports just mentioned only antimicrobial resistance was associated with treatment failure.
We (Mégraud F) reviewed the clinical relevance in 1998 at a time when data on susceptibility testing were scarce.92 One year later a systematic review was published based on data from 16 of 172 arms where proton pump inhibitor-clarithromycin-amoxicillin therapy was given, and for which susceptibility testing was performed.93 Table 4â–¶ is not a systematic review but presents data from 20 recent studies (1999–2003) and 1975 patients receiving the same treatment where susceptibility testing was performed. Unfortunately, these studies still represent a low proportion of the studies carried out (for example, 436 of 2751 patients among the French studies53) and sometimes the data are not presented in an adequate manner. A major difference in eradication rates was found: 87.8% when strains were clarithromycin susceptible compared with 18.3% when strains were clarithromycin resistant. The Mantel-Haenszel pooled odds ratio (OR) was highly significant (OR 24.5 (95% CI 17.2–35.0), p<0.001). It was 22.5 using the fixed effect model and 28.7 with the random effect model.
This 70% decrease in clinical success is higher than the 53% decrease reported in the meta-analysis by Houben and colleagues93 and confirms the high clinical relevance of clarithromycin resistance. Fortunately, the prevalence of clarithromycin resistance is still low in many places, as shown in table 1â–¶, but deserves monitoring at least in some centres.
Conversely, these results indicate the high success rate of this treatment (87.8%) when strains are susceptible, regardless of the proton pump inhibitor, dosage of the different drugs, and duration of the treatment.
Data from eight studies where a nitroimidazole compound was administered with clarithromycin instead of amoxicillin were also reviewed (table 5â–¶). When considering nitroimidazole resistance alone, there was a 25% decrease in the success rate compared with nitroimidazole susceptible strains (72.6% v 97%), which is identical to that observed by Houben and colleagues93 indicating that nitroimidazole resistance is less clinically relevant than clarithromycin resistance. The Mantel-Haenszel pooled OR was 11.3 (95% CI 5.7–22.3; p<0.001). It was 10.4 using the fixed effect model and 9.8 with the random effect model.
Furthermore, in the context of this treatment, clarithromycin resistance seems to lead less often to treatment failure (50% v 18.3%). A small number of strains were resistant to both antibiotics and none could be eradicated, reinforcing the fear of using this combination as a first line treatment because resistance to both drugs may be selected.
Few studies have used ranitidine bismuth citrate (RBC) instead of a proton pump inhibitor but data available indicate better efficacy of this combination, especially on metronidazole resistant strains (table 6â–¶). This result may be due to synergism between RBC and antibiotics.118 Such a synergy may also exist when a bismuth based quadruple therapy is used.26,119
The combination of proton pump inhibitor-amoxicillin-metronidazole has also been used in six trials (table 7â–¶). For metronidazole susceptible strains, the eradication rate was similar to the association of amoxicillin-clarithromycin (susceptible strains: 89.4% v 87.8%) which is inferior to the combination with clarithromycin (when strains are also susceptible: 89.4% v 97%; (p<0.001). A 25% drop in efficacy was also observed when strains were metronidazole resistant.
Interestingly, some authors compared a susceptibility testing strategy with an empirical treatment (table 8â–¶). There was a marked Sadvantage in using the susceptibility testing strategy in two studies. Indeed, the strategy depends very much on the treatment used. For Neri et al, an empirical treatment using RBC was the most efficient.125
Reference:
- Mégraud F. H pylori antibiotic resistance: prevalence, importance, and advances in testing. Gut. 2004 Sep; 53(9): 1374–1384. doi: 10.1136/gut.2003.0221112.
- Argueta, E.A.; Ho, J.J.C.; Elfanagely, Y.; D’Agata, E.; Moss, S.F. Clinical Implication of Drug Resistance for H. pylori Management. Antibiotics 2022, 11, 1684 https:// doi.org/10.3390/antibiotics11121684
Question 2. Where the resistance become more important?
Response 2 We have now replied Reviewer 1 comments and added the messages at line 398-402.
Submission Date : Response to Reviewer 1 Comments
6 March 2023

Reviewer 2 Report
The review deals with an important and interesting issue.
There are several long sentences which are difficult to read- for example line128-132.
Line 100- after "were reported" there should be "respectively"
Author Response
Point 1: There are several long sentences which are difficult to read- for example line128-132.
Response 1: We have re-edited long sentences to shorter (line 128-131) as requested by the reviewer.
Point 2: Line 100- after "were reported" there should be "respectively
Response 2 : We have already re-edited “respectively” as requested by the reviewer
Submission Date : Response to Reviewer 1 Comments
19 Feb 2023

Round 2
Reviewer 2 Report
The manuscript can be accepted
Author Response
Response to Reviewer 2 Comments : 2th Revision
Point 1: There are several long sentences which are difficult to read
Response 1: We have re-edited long sentences to shorter (line 128-131) as requested by the reviewer.
Submission Date : Response to Reviewer 1 Comments
6 March 2023
